SARS-CoV-2 ORF3A interacts with the Clic-like chloride channel-1 (CLCC1) and triggers an unfolded protein response

Gruner Hannah N. 1
Zhang Yaohuan 1 2
Shariati Kaavian 1
Yiv Nicholas 1
Hu Zicheng 3
Wang Yuhao 1
Hejtmancik J. Fielding 4
McManus Michael T. 1
Tharp Kevin 5
Ku Gregory gregory.ku@ucsf.edu 1 6
1 Diabetes Center, University of California , San Francisco , CA , United States of America
2 Metabolic Biology Graduate Program, University of California , Berkeley , CA , United States of America
3 Bakar Computational Health Sciences Institute, University of California , San Francisco , CA , United States of America
4 National Eye Institute , Bethesda , MD , United States of America
5 Center for Bioengineering and Tissue Regeneration, University of California , San Francisco , CA , United States of America
6 Division of Endocrinology and Metabolism, University of California , San Francisco , CA , United States of America
Silva Pedro
Electronic publication date: 2023 Apr 3
Publication date: 2023
Volume: 11
Electronic Location ID: e15077
Received 2022 Oct 14; Accepted 2023 Feb 24
Copyright: ©2023 Gruner et al.
Copyright year: 2023
Copyright holder: Gruner et al.
License: This is an open access article distributed under the terms of the Creative Commons Attribution License, which permits unrestricted use, distribution, reproduction and adaptation in any medium and for any purpose provided that it is properly attributed. For attribution, the original author(s), title, publication source (PeerJ) and either DOI or URL of the article must be cited.
License URL: https://creativecommons.org/licenses/by/4.0/

Keywords: COVID-19, SARS-CoV-2, CLCC1, Unfolded protein response

Funding: UCSF COVID-19 Rapid Response Pilot Grant DK107650 DK118337 This work was supported by the UCSF COVID-19 Rapid Response Pilot Grant, NIH grants DK107650 and DK118337 to Gregory Ku and 1U24DK116214-01 to Michael McManus. The funders had no role in study design, data collection and analysis, decision to publish, or preparation of the manuscript.

==============================
Understanding the interactions between SARS-CoV-2 and host cell machinery may reveal new targets to treat COVID-19. We focused on an interaction between the SARS-CoV-2 ORF3A accessory protein and the CLIC-like chloride channel-1 (CLCC1). We found that ORF3A partially co-localized with CLCC1 and that ORF3A and CLCC1 could be co-immunoprecipitated. Since CLCC1 plays a role in the unfolded protein response (UPR), we hypothesized that ORF3A may also play a role in the UPR. Indeed, ORF3A expression triggered a transcriptional UPR that was similar to knockdown of CLCC1. ORF3A expression in 293T cells induced cell death and this was rescued by the chemical chaperone taurodeoxycholic acid (TUDCA). Cells with CLCC1 knockdown were partially protected from ORF3A-mediated cell death. CLCC1 knockdown upregulated several of the homeostatic UPR targets induced by ORF3A expression, including HSPA6 and spliced XBP1, and these were not further upregulated by ORF3A. Our data suggest a model where CLCC1 silencing triggers a homeostatic UPR that prevents cell death due to ORF3A expression.

Introduction

A greater understanding the host-pathogen interactions of SARS-CoV-2, the virus that causes COVID-19, may reveal novel therapeutic targets to reduce the global impact of this disease. While appropriate attention has been paid to the interaction of the spike protein with the ACE2 and TMPRSS2 proteins, the accessory proteins of SARS-CoV-2 are also important for its pathogenesis. SARS-CoV-2 ORF3A is the largest of the accessory proteins, with 275 amino acids encoding a 3-pass transmembrane protein. SARS-CoV-2 ORF3A is 73% identical to SARS-CoV ORF3A, suggesting that insights into SARS-CoV ORF3A may be relevant to the study of SARS-CoV-2 ORF3A. A SARS-CoV virus with a deletion of ORF3A is less cytotoxic without affecting viral replication (Freundt et al., 2010). Expression of SARS-CoV ORF3A alone induces golgi fragmentation (Freundt et al., 2010), the unfolded protein response (UPR) (Minakshi et al., 2009) and NLRP3 inflammasome activation, and cell death (Siu et al., 2019).

Like its counterpart in SARS-CoV, SARS-CoV-2 ORF3A induces cell death (Ren et al., 2020). Point mutations in either the cysteine-rich motif or the YXX Φ motif in SARS-CoV-2 ORF3A reduce membrane localization and reduce its ability to trigger cell death (Ren et al., 2020). Deletion of ORF3A from SARS-CoV-2 substantially reduces its pathogenicity in human ACE2 transgenic mice (Silvas et al., 2021). The cryo-electron microscopy structure of ORF3A reveals a dimeric or tetrameric channel and purified ORF3A in nano disks exhibits non-selective cation channel activity (Kern et al., 2020). SARS-CoV-2 ORF3A interacts with a wide variety of different host proteins and processes (Gordon et al., 2020). SARS-CoV-2 ORF3A-induced cell death is blocked by caspase-8 and caspase-9 inhibitors (Ren et al., 2020). It sequesters VPS39 and prevents autolysosome formation (Miao et al., 2021). ORF3A also localizes to late endosomes; alters endosome morphology (Miserey-Lenkei et al., 2021); and binds to STING to block NF-kB activation (Rui et al., 2021).

High throughput protein interaction studies of SARS-CoV-2 proteins identified the CLIC-like chloride channel-1 (CLCC1) as an ORF3A interacting protein (Gordon et al., 2020) and a protein that is differentially phosphorylated upon SARS-CoV-2 infection (Bouhaddou et al., 2020). CLCC1 is a putative chloride channel that localizes predominantly to the endoplasmic reticulum (Nagasawa et al., 2001). A spontaneous mutation disrupting the Clcc1 gene in mice leads to endoplasmic reticulum (ER) stress and cell death in the cerebellum, resulting in ataxia (Jia et al., 2015). In humans, a point mutation in CLCC1 causes autosomal recessive retinitis pigmentosa likely due to increased retinal cell death, presumably from increased ER stress (Li et al., 2018).

Given that expression of SARS-CoV ORF3A is sufficient to induce the UPR (Minakshi et al., 2009), and recent studies have suggested SARS-Cov-2 elicits a UPR similar to other coronaviruses (Balakrishnan & Lai, 2021; Bartolini et al., 2022; Rosa-Fernandes et al., 2021; Shaban et al., 2021), and that CLCC1 plays a role in ER stress (Jia et al., 2015; Li et al., 2018), we explored a functional link between CLCC1 and SARS-CoV-2 ORF3A. We confirmed that ORF3A and CLCC1 physically interact. ORF3A expression induced a homeostatic unfolded protein response as did CLCC1 silencing. Our data suggest that ORF3A triggers the UPR through CLCC1 and this may play a role in SARS-CoV-2 induced cytotoxicity.

Materials & Methods

Molecular biology

pDONR207 SARS-CoV-2 ORF3A was a gift from Fritz Roth (Addgene plasmid #141271; http://n2t.net/addgene:141271; RRID:Addgene_141271). An HA tag was added to the 3′ end of the open reading frame by PCR and was cloned into pSicoR EF1a, where the EF1a promoter directs expression of ORF3A-HA. The correct sequence was validated by Sanger sequencing. Two guide RNAs to silence human CLCC1 were cloned into a lentivirus vector expressing from the U6 promoter with GFP and hygromycin resistance. The guide RNA sequences were: GGGAAGCACGCTGAAACCCT and GCCCAGGCCGGCCGCAGAAG. After lentiviral infection of 293T cells stably expressing dCas9KRAB, cells were selected for lentiviral infection using hygromycin (Thermo).

The wild type and human CLCC1 and D25E mutant CLCC1 with a N-terminal FLAG tag were subcloned by PCR into a lentiviral expression vector downstream of mCherry-T2A, driven by the CMV promoter. The correct sequence was validated by Sanger sequencing.

Cells

293T cells stably expressing dCas9KRAB were generated by integration of a donor cassette containing a CMV promoter driving dCas9KRAB with separate EF1a promoter driving puromycin and mCherry flanked by loxP sites. The parental 293T cell was obtained from ATCC. Eight hundred based pair homology arms to the AAVS1 locus flanked the cassettes. The donor DNA was co-transfected with a wild type cDNA and a guide RNA targeting the AAVS1 genomic locus (GGGGCCACTAGGGACAGGAT) (Mali et al., 2013). Clones were selected using puromycin. Homologous integration was confirmed by genomic DNA PCR. To remove the puromycin and mCherry, 800ng of the Cre plasmid (Addgene plasmid #24971) was transfected and the bulk population was serially diluted to generate single cell clones. Each clone was validated for mCherry loss and puromycin sensitivity and then silencing was validated using silencing of CD146 as a test target.

mRNA-seq

Total RNA was isolated using an RNeasy Plus Mini kit (Qiagen). PolyA RNA was isolated from one microgram of total RNA (Lexogen, Vienna, Austria) and stranded RNA-seq libraries were prepared (CORALL; Lexogen, Vienna, Austria), indexed and sequenced on a HiSeq 4000 (Illumina, San Diego, CA, USA) at the UCSF Center for Advanced Technologies. Reads were aligned by HiSat2 (Pertea et al., 2016) to the hg38 assembly and gene level counts were made using HTSeq (Anders, Pyl & Huber, 2015). The statistical power of this design is 0.75 given a sequencing depth of 4, 3 biological replicates, 0.1 coefficient of variation, to detect a 3-fold difference with an alpha value of 0.05 (Hart et al., 2013).

Differential expression analysis were performed with DESeq2 (Love, Huber & Anders, 2014). The interaction between CLCC1 and ORF3A were tested with DESeq2 with the following model: Count ∼ CLCC1 + ORF3A + CLCC1 * ORF3A. GO term enrichment was performed using the Gene Ontology website (release date = 2020-07-16) (Ashburner et al., 2000; Mi et al., 2019; The Gene Ontology , 2019). Spliced XBP1 mRNA was calculated as a percent of total XBP1 by adding the transcripts per million of ENST00000344347.5 + ENST00000611155.4 divided by ENST00000216037.10 + ENST00000344347.5 + ENST00000611155.4. Significance was determined by ratio ∼ CLCC1 + ORF3A + CLCC1*ORF3A.

Sequences have been deposited at the NCBI Sequence Read Archive (https://www.ncbi.nlm.nih.gov/sra) under accession number PRJNA887134.

Cell death assay—flow cytometry

A total of 48 h after transient transfection with plasmids (Jetprime), cells were trypsinized and incubated with Sytox Orange (Thermo Fisher, Waltham, MA, USA) to label dead cells, then analyzed by flow cytometry (Attune; Thermo Fisher Scientific). Forward and side scatter gates were used to identify for intact cells and the % of cells that were Sytox Orange positive was determined.

Cell death assay—incucyte

Cells were transiently transfected (Jetprime) with the indicated plasmids into 96 well plates. To reduce cell toxicity, the media was changed 4 h after transfection and 15 nM final Sytox Green (Thermo Fisher Scientific) was added to allow detection of dead cells. The % of dead cells was calculated as the number of Sytox Green positive nuclei that overlapped mCherry positive cells divided by the total number of mCherry positive cells. Images were taken 36 h post-transfection.

Immunoprecipitation

48 h after transient transfection, cells were lysed in buffer containing 20 mM Tris pH 7.5, 150 mM NaCl, 1% Triton X-100, and 1x cOmplete protease inhibitor cocktail without EDTA (Roche, Basel, Switzerland). Lysates were cleared by centrifugation at 21,000xg for 10 min and the supernatant was incubated with 1-5 micrograms of antibody pre-bound to 100 uL of protein G Surebeads (Bio-Rad, Hercules, CA, USA). After tumbling for 2 h at 4 degrees C, the beads were washed four times with 300 uL of lysis buffer, eluted with 1x LDS sample buffer and western blots were performed.

Antibodies

M2 anti-FLAG (1:100 immunofluorescence, 1:1000 western blot, Sigma F1804), 3F10 anti-HA (1:100 immunofluorescence, 1:3000 western blot; Roche), anti-HA-Tag-488 (1:200; Santa Cruz, sc-7392), anti-GAPDH-hrp (1:10,000 western blot, Sigma, G9525), anti-CLCC1 (1:500, western blot, 26680-1-AP; Proteintech), anti-CLCC1 (1:50, immunofluorescence, Novus Biologicals, NBP1-82793), anti-PDI (1:100 immunofluorescence, 45596; Cell Signaling Technology), anti-PDI (1:100, ADI-SPA-891; Enzo) conjugated to Alexa Fluor™ 647 with the Alexa Fluor™ 647 Protein Labeling Kit (Thermofisher, A20173), anti-EEA1 (1:100 immunofluorescence, 66218-1-Ig; Proteintech), anti-RCAS1 (1:100, 12290; Cell Signaling Technology), wheat germ agglutinin (Invitrogen, WGA-555 or WGA-488 5.0 µg/mL), Hoechst 33342 (Thermo Fisher Scientific 1:2000).

Microscopy

Cells were grown on PerkinElmer CellCarrier-96 ultra tissue culture treated microplates (PerkinElmer, Waltham, MA, USA). Growth media was removed and cells were fixed with freshly prepared 4% PFA in PBS for 10 min at room temperature and then washed in 1X PBS. Cells were first assessed to confirm transfection success through visualizing mCherry or GFP. Cells were permeabilized and blocked in PBST (1x PBS; 0.1% Triton X-100, 10% FBS) for 90 min at room temperature. Fluorophore signal was inactivated using Cyclic Immunofluorescence (PMID: 27925668). Cells were imaged on an Opera Phenix Spinning Disk Confocal (PerkinElmer). A single optical slice is shown.

Image analysis

ORF3A intensity analysis. Using Harmony image analysis software, segmentation was performed on the nucleus, cell, cytoplasm, membrane for all cells. Cells touching the border were excluded. Within the cell mask, various organelles were segmented depending on the markers used which included ER (PDI), golgi (RCAS1), lysosome (LAMP1), early endosomes (EEA1). The median intensity for anti-HA-ORF3A staining was calculated within the subcellular compartment masks for >1,400 ORF3A transfected cells for each organelle. Within cells, regions with and without ORF3A staining were segmented by using non-transfected control wells to determine thresholds.

Results

A high throughput SARS-CoV-2 protein interaction map identified CLCC1 as an ORF3A interacting protein (Gordon et al., 2020). Confirming these high throughput data, ORF3A immunoprecipitates contained endogenous CLCC1 (Fig. 1A). Reciprocally, immunoprecipitates of FLAG tagged, wild type CLCC1 contained ORF3A (Fig. 1B). We also tested the ability of a hypomorphic CLCC1 disease-associated variant with reduced channel activity, CLCC1 D25E (Li et al., 2018), and found this mutation did not affect CLCC1 binding to ORF3A (Fig. 1B). In cells, endogenous CLCC1 co-localized with the ER as has been previously reported (Li et al., 2018) and a minority of ORF3A-HA (Fig. 1C). FLAG-CLCC1 or FLAG-CLCC1 D25E again co-localized predominantly with the ER, and with minority of ORF3A-HA (Fig. 1D). The majority of ORF3A did not colocalize with ER, golgi (Fig. 1E), plasma membrane (Fig. 1F), or lysosome (Fig. 1G) and instead colocalized with the early endosome marker EEA1 (Fig. 1G). The median intensity of ORF3A staining was highest in the early endosome (Fig. 1H).

Figure 1 CLCC1 and ORF3A physically interact.

293T cells were transfected with the indicated plasmids (top). Forty-eight hours after transfection, the cells were lysed and immunoprecipiataion was performed with the indicated antibody. (A) Western blots for CLCC1 and HA are shown for the anti-HA immunoprecipitation. (B) Western blots for HA and FLAG are shown for the anti-FLAG precipitation. Both panels representative of three independent experiments. (C) 293T cells transfected with the indicated plasmids were stained for endogenous CLCC1, ORF3A (anti-HA) and the endoplasmic reticulum (anti-PDI) n = 2. (D) 293T cells were transfected with wild type CLCC1 of D25E CLCC1 and ORF3A-HA and stained for FLAG, HA, and PDI n = 3. (E) 293T cells were transfected with ORF3A-HA and stained for PDI and RCAS1 (golgi) n = 3. (F) 293T cells were transfected with ORF3A-HA and stained for HA and WGA (plasma membrane) n = 4. (G) 293T cells were transfected with ORF3A-HA and stained for HA, EEA1, and LAMP1 n = 4. (H) For each marker, the median ORF3A intensity in the listed organelle is plotted. Standard deviation is shown. The blue dashed line indicates the median intensity of anti-HA-ORF3A for all ORF3A positive masks, while the red dashed line represents the intensity outside the ORF3A mask within cells. n = 4 lysosome, golgi, ER, endo-some, n = 8 whole cell, nucleus, membrane, cytoplasm.

To ask if ORF3A expression results in a transcriptional unfolded protein response (UPR), we examined the global transcriptional response to SARS-CoV-2 ORF3A expression using bulk mRNA-sequencing. ORF3A upregulated 21 genes after multiple testing correction (Fig. 2A and Table S1). Nearly all GO terms enriched in these 21 genes were related the response to unfolded proteins (Fig. 2B). The most highly upregulated gene (>100 fold) was HSPA6, an HSP70 member chaperone classically induced by heat shock, unfolded proteins and dysregulation of HSP90 (Deane & Brown, 2018; Kuballa et al., 2015). However, other unfolded protein response (UPR) genes were also upregulated, including HERPUD1, a component of the ER associated protein degradation machinery (Belal et al., 2012); XBP1, a master regulator of the UPR (Oakes & Papa, 2014); CTH, the cystathionine gamma-lyase (Maclean et al., 2012); and ASNS, asparagine synthetase (Lomelino et al., 2017).

Figure 2 ORF3A expression and CLCC1 silencing both trigger the unfolded protein response.

(A) Volcano plot of gene expression after ORF3A expression compared to control (adjusted p value < 0.05, fold change >1.9 in green). (B) GO terms enriched in genes that are statistically significantly upregulated (adjusted p value < 0.05) after ORF3A expression. (C) Volcano plot of gene expression after CLCC1 silencing expression compared to control (adjusted p value < 0.05, fold change > 1.9 in green). (D) GO terms enriched in genes that are statistically significantly upregulated (adjusted p value < 0.05) after CLCC1 silencing.

Interleukin 1-beta, which is upregulated by SARS-CoV ORF3A expression in 293T cells (Siu et al., 2019), was not induced by SARS-CoV-2 ORF3A expression (no counts in either condition, File S1). Only six genes were downregulated after ORF3A expression. Though there was no statistically significant enrichment of any GO term in the downregulated gene list, we noted that TXNIP, a pro-apoptotic UPR gene (Lerner et al., 2012; Oslowski et al., 2012), was downregulated by ORF3A expression while the upregulated UPR genes (HSPA6, HERPUD1, XBP1, CTH, ASNS) are predicted to increase protein folding capacity. DDIT3, another pro-apoptotic UPR gene, was not changed by ORF3A expression (Table S1). Notably, upregulation of HSPA6 and downregulation of TXNIP has been reported in 293T cells after SARS-CoV-2 infection (Sun et al., 2021).

We then examined gene expression after knockdown of CLCC1 using CRISPR interference. CLCC1 knockdown caused upregulation of nine genes (Fig. 2C and Table S2). Consistent with prior data on CLCC1, the upregulated genes were highly enriched for GO terms involving the response to unfolded proteins (Fig. 2D). Interestingly, as in ORF3A expressing cells, the most highly upregulated gene after CLCC1 silencing was HSPA6, showing a 50-fold increased mRNA expression. HERPUD1, upregulated by ORF3A expression, was also upregulated by CLCC1 silencing. Other UPR genes were upregulated in CLCC1 silenced cells that were not increased in ORF3A expressing cells, including DNAJB1, HSPA5, and HYOU1. There were only 4 genes that showed downregulation after CLCC1 silencing (besides CLCC1 itself) and no GO term was enriched.

Like SARS-CoV ORF3A (Law et al., 2005), SARS-CoV-2 ORF3A expression induced cell death as has been previously reported (Ren et al., 2020) (Figs. 3A–3B). SARS-CoV-2 ORF3A induced cell death was blocked by incubation with the chemical chaperone taurodeoxycholic acid (TUDCA) (Figs. 3C–3D), suggesting a that ORF3A induced cell death might involve ER stress.

Figure 3 ORF3A expression induces cell death that is blocked by TUDCA.

(A) 293T cells were transfected with either a vector expressing mCherry or a vector expressing ORF3A-HA. The fraction of cells that were Sytox Green at 48 hours by flow cytometry is plotted. n = 6, ** p < 0.01 by Student’s t-test. (B) Cells from A were lysed and a western blot was performed for HA (top) or GAPDH (bottom). (C) As in A, but cells were pre-treated with TUDCA at the indicated concentrations. (D) Cells from C were lysed and a western blot was performed for HA (top) or vinculin (bottom) or CLCC1 (middle).

We then asked if CLCC1 knockdown would impact ORF3A-mediated cell toxicity. Cells with knockdown of CLCC1 were modestly less susceptible to cell death in response to ORF3A over-expression (Fig. 4A) despite comparable levels of ORF3A expression (Fig. 4B). Over-expression of wild type CLCC1 alone did not induce cell death. However, wild type CLCC1 over-expression synergized with ORF3A expression to induce cell death. In contrast, over-expression of the D25E mutant allele of CLCC1 alone caused cell death and this was marginally increased after ORF3A expression (Fig. 4C). Levels of ORF3A were similar between all conditions of CLCC1 transfection and control DNA transfection (Fig. 4D).

Figure 4 CLCC1 functionally interacts with ORF3A.

(A) 293T cells expressing the indicated guide RNAs (top) were transfected with either mCherry or ORF3A-HA. Thirty-six hours after transfection, cell death was normalized to the % cell death observed in non-targeting sgRNA expressing cells with ORF3A expression n = 11. Boxes indicate the median and 75% and 25% percentiles. Two-way ANOVA showed a significant interaction between CLCC1 knockdown and ORF3A over-expression F(2, 48) = 25.82 p = 2.43E − 8. Tukey’s HSD post-hoc family wise comparison of means: **** p < 0.0001. Significance for all comparisons is shown in File S3. (B) As in A, but western blots for CLCC1 (top panel), ORF3A-HA (middle panel), and GAPDH (bottom panel). (C) 293T cells were transfect-ed with the indicated plasmids and cell death was measured at 36 h. n = 10 for each condition. Two-way ANOVA showed a significant interaction between ORF3A expression and CLCC1 over-expression: F(2, 54) = 31, 77, p = 7.59e − 10. (D) As in D, but western blots were performed for CLCC1 (bottom panel), ORF3A (top panel), and GAPDH (middle panel). (E) Variance stabilizing transformed counts for the indicated genes under the indicated conditions from mRNA-seq. (F) Fraction of spliced XBP1 under indicated conditions.

Since CLCC1 knockdown activated a homeostatic UPR (upregulating genes such as HSPA5, HSPA6, HERPUD1, HYOU1, DNAJB1), we hypothesized that these CLCC1 knockdown cells might be better prepared for the UPR stress of expression of ORF3A. We examined gene expression in CLCC1-silenced cells after ORF3A expression and tested for genes regulated by the interaction of CLCC1 and ORF3A. Two genes were regulated by the interaction of CLCC1 and ORF3A—HSPA6 and HERPUD1 (FDR corrected p values of 8.22e−12 and 8.1e−02 respectively). These genes were upregulated by CLCC1 knockdown and were only modestly further upregulated after ORF3A expression (Fig. 4E).

Given the upregulation of XBP1 by ORF3A expression, we examined the abundance of the spliced form of XBP1 (sXBP1) that is generated by activated IRE1. It is this transcript of XBP1 that encodes a transcription factor that drives the homeostatic UPR (Calfon et al., 2002; Yoshida et al., 2001). There was a statistically significant interaction between CLCC1 knockdown and ORF3A on sXBP1 abundance (p = 0.04). ORF3A caused an increase in sXBP1 (p = 0.01) in cells with normal CLCC1 expression, but did not do so in cells that had CLCC1 knockdown (p = 0.87). CLCC1 knockdown alone caused an increase in spliced XBP1 expression that did not reach statistical significance (p = 0.0786) (Fig. 4F), showing another homeostatic UPR gene that was not further upregulated by ORF3A expression in the setting of CLCC1 knockdown.

Discussion

Understanding host pathogen interactions for SARS-CoV-2 may lead to new therapeutic opportunities to treat COVID-19. We confirmed high throughput protein interaction data showing that ORF3A physically interacts with the CLCC1, a protein which is required for maintenance of ER homeostasis in retinal cells and in cerebellar granule cells (Jia et al., 2015; Li et al., 2018). Our data show that CLCC1 and ORF3A also may functionally interact—the knockdown or over-expression of CLCC1 impacted ORF3A-induced cell toxicity and the upregulation of UPR genes by ORF3A was weaker in CLCC1 knockdown cells.

From these data, however, we cannot confidently conclude the molecular mechanism of CLCC1 and ORF3A interaction or if there is indeed a functional interaction. One model is that ORF3A inactivates CLCC1 through direct or indirect action, leading to ER stress and cell death. However, we do not see increased cell death with CLCC1 knockdown. Additionally, TUDCA relieves cell death due to ORF3A. It is possible the cation channel activity of ORF3A influences CLCC1 function or directly triggers an inflammatory response, but the role of the SARS-CoV-2 ORF3A acting as an ion channel is only beginning to emerge (Kern et al., 2020). Together these data suggest that ORF3A triggers ER stress and cell death independent of CLCC1. Indeed, ORF3A triggers additional pathways independent of ER stress that lead to cell death (e.g., caspase-8 and caspase-9).

We favor the model that ORF3A inactivation of CLCC1 does not increase cell death, but actually triggers a homeostatic UPR. Indeed, terminal UPR genes were not upregulated by CLCC1 knockdown, but homeostatic UPR genes that are predicted to increase folding capacity were (e.g., HSPA6, HERPUD1). In this model, knockdown of CLCC1 might be predicted to reduce ORF3A toxicity by increasing this protective homeostatic UPR response. On the other hand, the over-expression of CLCC1 resulted in increased cell death, which may suggest CLCC1 overexpression might overcome CLCC1 inactivation by ORF3A, preventing this homeostatic UPR normally caused by potential CLCC1 suppression. The over-expression of CLCC1 D25E did not result in increase cell death to the same level as wild type CLCC1 when over-expressed with ORF3A, suggesting that functional CLCC1 is important for the synergy with ORF3A. However, we did note increased cell death with CLCC1 D25E over-expression alone, suggesting that these D25E over-expression experiments be interpreted with caution. Without reciprocal mutations and binding studies, we cannot rule out an effect of CLCC1 over-expression or knockdown that is independent of its binding to ORF3A or even the UPR, especially given the multiple activities of ORF3A. For example, ORF3A inhibits autophagy (Miao et al., 2021), which normally promotes cell survival during chronic ER stress (Ogata et al., 2006). Thus, it is plausible ORF3A overexpression cells are more susceptible to ER stress mediated cell death due to inhibition of the protective autophagy pathway.

ORF3A interacts with many other host proteins besides CLCC1. We suggest that another leading candidate is HMOX1, the heme oxygenase-1, which normally serves to reduce oxidative stress. We found that HMOX1 mRNA is reduced by ORF3A expression and HMOX1 also physically interacts with ORF3A (Gordon et al., 2020). These data suggest ORF3A interferes with HMOX1 contributions to the oxidative stress response, as others have suggested (Batra et al., 2020).

In contrast to our findings in 293T cells where CLCC1 silencing did not cause cell death, CLCC1 depletion in retinal cells and cerebellar cells caused cell death that is presumed to be due to a terminal UPR (Jia et al., 2015; Li et al., 2018). This difference may be because retinal and neuronal cell types are more susceptible to ER stress induced cell death. Therefore, depending on the cell type infected with SARS-CoV-2, the interaction of ORF3A and CLCC1 may have different outcomes. We also note that ORF3A-HA localized to the EEA1 + early endosome in our studies and not the late endosome as other authors have observed (Miserey-Lenkei et al., 2021). This may be due to the level of over-expression of ORF3A or to differences in the cells used. Our study is an in vitro study using a recombinant expression system and should be interpreted as such. Most importantly, we studied ORF3A function in isolation (out of the context of virus infection).

Conclusions

We conclude that SARS-CoV-2 ORF3A activates an UPR and may functionally interact with host CLCC1. Modulation of the CLCC1 activity and the UPR might reduce cytotoxicity from ORF3A in the setting of SARS CoV-2 infection.

Supplemental Information

File S1 Differentially expressed genes after ORF3A over-expression

DESeq2 output: mCherry over-expression versus ORF3A-HA overexpression.

Click here for additional data file.

File S2 Differentially expressed genes after CLCC1 knockdown

DESeq2 output comparing control knockdown and CLCC1 knockdown.

Click here for additional data file.

File S3 Post-hoc testing statistics for Figure 4A and 4C

Click here for additional data file.

Supplemental Information 1 Uncropped western blots

Click here for additional data file.

Data S1 Raw data

Click here for additional data file.

Additional Information and Declarations

Competing Interests

Author Contributions

DNA Deposition

Data Availability

The authors declare there are no competing interests.

Hannah N. Gruner conceived and designed the experiments, performed the experiments, analyzed the data, prepared figures and/or tables, authored or reviewed drafts of the article, and approved the final draft.

Yaohuan Zhang conceived and designed the experiments, performed the experiments, analyzed the data, prepared figures and/or tables, authored or reviewed drafts of the article, and approved the final draft.

Kaavian Shariati performed the experiments, authored or reviewed drafts of the article, and approved the final draft.

Nicholas Yiv performed the experiments, authored or reviewed drafts of the article, and approved the final draft.

Zicheng Hu performed the experiments, analyzed the data, authored or reviewed drafts of the article, and approved the final draft.

Yuhao Wang performed the experiments, authored or reviewed drafts of the article, and approved the final draft.

J. Fielding Hejtmancik conceived and designed the experiments, authored or reviewed drafts of the article, and approved the final draft.

Michael T. McManus conceived and designed the experiments, analyzed the data, authored or reviewed drafts of the article, and approved the final draft.

Kevin Tharp conceived and designed the experiments, analyzed the data, authored or reviewed drafts of the article, and approved the final draft.

Gregory Ku conceived and designed the experiments, analyzed the data, prepared figures and/or tables, authored or reviewed drafts of the article, and approved the final draft.

The following information was supplied regarding the deposition of DNA sequences:

RNA-seq data have been uploaded to the Sequence Read Archive (NCBI) under accession number PRJNA887134.

The following information was supplied regarding data availability:

The uncropped western blots are available in the Supplemental File.

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
