# Peer review of "SARS-CoV-2 ORF3A interacts with the Clic-like chloride channel-1 (CLCC1) and triggers an unfolded protein response"

_PeerJ, doi:10.7717/peerj.15077_

## Round 0.1 · original submission · Major Revisions

We have obtained three in-depth reviews of your work. Two of them are very positive, but one reviewer has doubts regarding the role of CLCC1 and the physiological relevance of the levels of TUDCA used. I look forward to receiving a revised manuscript.

Reviewer 1 ·

Basic reporting

no comment

Experimental design

no comment

Validity of the findings

This work begins to examine a functional interaction between the SARS-CoV-2 accessory protein Orf3a and host protein CLCC1, which the authors ascribe to the endoplasmic reticulum (ER) homeostatic unfolded protein response (UPR). SARS-CoV Orf3a had been reported to induce UPR and disruptions to CLCC1 may be associated elevated ER stress and cell death. A number of studies investigating the interaction of host proteins with individual SARS-CoV-2 and SARS-CoV proteins have identified CLCC1 as a putative, yet conserved, interacting partner of Orf3a proteins. This suggests a common mechanism of action shared by Orf3a proteins and CLCC1 that has yet to be elucidated. Although Orf3a proteins have been characterized as putative viroporins, recent work has challenged this hypothesis. Advancing our knowledge of viral pathogenesis is an important area of COVID-19 research and is a difficult, yet critical question.

In this manuscript, the authors validate that CLCC1 and SARS-CoV-2 Orf3a interact by co-immunoprecipitation and show that the two proteins, in part, co-localize in HEK293 cells. They next ask which genes are differentially up- or down-regulated upon 1) SARS-CoV-2 Orf3a overexpression and 2) CLCC1 depletion. They identify several genes that similarly up-regulated in both datasets that are implicated in the homeostatic UPR pathway – HSPA6 and HERPUD1. They demonstrate that Orf3a overexpression induces cell death and claim that this is be UPR-mediated since it can be curtailed with the addition of TUDCA. And finally, they investigate whether CLCC1 and SARS-CoV-2 Orf3a may function in synergy by evaluating toxicity in cells that are devoid of, or are enriched with, CLCC1 in the presence of Orf3a. They observe a modest reduction in death of cells devoid of CLCC1. CLCC1 knockdown is hypothesized by the authors to prime cells for homeostatic URP and would to protect cells from SARS-CoV-2 Orf3a-mediated cell death. They also observe enhanced cell death when both protein are expressed. However, cell death was not associated with an increase in transcript levels for UPR pathway genes HSPA6, HERPUD1 and the splice form of XBP1.

Major comments:

Although the authors are careful about the discussion of their data, my overall fear is that, at present, their work does not support a functional interaction between SARS-CoV-2 Orf3a and CLCC1. This is in part because the physiological role CLCC1 is not clear. It has been annotated a chloride channel, but the evidence for this is weak. It may be an ER-resident trafficking protein, but mechanistically this is not well understood. The most convincing data presented here are 1) the biochemical interaction of SARS-CoV-2 Orf3a and CLCC1, 2) the upregulation of genes in the UPR pathway by both Orf3a overexpression and CLCC1 knockdown, and 3) the enhanced cell toxicity observed with Orf3a overexpression. The data supporting the idea that cell death by Orf3a overexpression is UPC-mediated uses very high concentrations (0.1-0.5 mM) of TUDCA and is therefore likely non-specific and not convincing currently. Apart from physical interaction (direct or indirect), none of these data supports a functional interaction between these two proteins, unfortunately. My best recommendation to the authors would be to step back and try an unbiased approach to establish the functional interaction between SARS-CoV-2 Orf3a and CLCC1. The interaction does appear to be conserved between CLCC1 and SARS-CoV or SARS-CoV-2 Orf3a, based on prior work and validated here. This suggests a functional importance to the interaction. However, with limited knowledge of CLCC1 function, this is extremely challenging question to address.

·

Basic reporting

Gruner et al present a study on the interaction of the ORF3a protein of SARS-CoV-2 with the CLCC1 intracellular chloride channel protein. Based on an interaction map of SARS-CoV-2 proteins, expression of HA-tagged ORF3a and interaction with CLCC1 was tested in HEK293T cells. Immunoprecipitaion experiments clearly show an interaction between CLCC1 and ORF3a, recombinantly expressed ORF3a was found to localize predominantly to late ensdosomes. Expression of ORF3a triggered increased expression of heat shock- and ER stress-related proteins, similar upregulation pattern was observed when CLCC1 was downregulated. ORF3a-induced cell death was reduced upon administration of taurodeoxycholic acid, a “chemical chaperone” known to reduce ER Stress and stabilizing the unfolded protein response. From their data, the authors conclude that the ORF3a – CLCC1 interaction may activate an unfolded protein response as part of ORF3a – induced cytotoxicity.

Experimental design

The paper is well written, the topic is introduced clearly, experiments are thoroughly done and Figures clear and to the point. The work is of excellent quality and certainly meeting the standards for an international reviewed publication in PeerJ.

Validity of the findings

The topic is of relevance, especially since a steady state of infection and prevalence of COVID19 has been reached that makes the search for effective therapies a very important field. Thus, the paper offers an important extension of knowledge on the pathomechanisms and therapy of SARS-CoV-2 associated disease.

Additional comments

The paper needs some very minor adjustments – on the discussion level only – before being acceptable for publication.

Recommendation: publish after minor correction. No need to peer review the modified manuscript again.

Points for consideration

1) It is interesting that both, ORF3a and CLCC1 are intracellular ion channels. There could be a possible interference from the concomitant action of an anion (CLCC1) and a cation (ORF3a) channel. While not focus of the study, the role of channel function – in addition to protein-protein interaction – should be mentioned in the discussion. Viroporins have been proposed to trigger inflammatory responses through their channel activity.
2) Figure 2E – correlation of gene induction upon CLCC1 k.o. vs. ORF3a expression. Given a correlation coefficient of R = 0.37, there appears to be little statistical significance in this correlation – especially when considering the logarithmic scales. The figure could be omitted. It is already discussed with caution in the present paper.
3) Line 322-323 “Our study has other several (severe??) limitations … “. Suggestion to omit the sentence on limitations, modesty is nice but no need to overdo it. IT is an in vitro study using a recombinant expression system, this could be mentioned. And the study does indeed demonstrate a novel interaction convincingly.

Reviewer 3 ·

Basic reporting

First of all, I would like to point out that the manuscript is very interesting.
I do believe -however- that there could be some rephrasing in order to make it more clear and unambiguous. Also, I'd clarify that the paper is always talking about the ORFa protein and not the gene.
In terms of context provided, I do believe that the introduction could touch more on the fact that SARS-Cov2- has been shown to trigger an UPR and include references in that sense.
The article is well-structured with proper figures. Raw data was shared accordingly.
The article is in fact self-contained and the results are relevant to its objectives.

Experimental design

The research presented in this manuscript was well within the Aims and Scope of this Journal. The investigation was performed rigorously.
Moreover, the research question is relevant and meaningful. I'd advise the authors to develop a bit more how their work contributes to add to the ongoing knowledge on SARS-CoV-2 cytotoxicity.
I'd also like to point out that the methods have been thoroughly described, with adequate details and info in order to replicate.

Validity of the findings

Once again, I'd like to point out that this manuscript and its findings are very interesting. The study presents robust data that support accordingly the original research question. The conclusions are adequate.

---

## Round 0.2 · accepted · Accept

Thank you for addressing our reviewers' requests and toning down some of the conclusions. I am glad to accept your manuscript for publication

·

Basic reporting

This is a revised version of the original manuscript.
My concerns were little for the original submission. Points raised by myself were answered satisfactorily.
The authors have followed suggestions to tone down some claims. In the present form I copnsider the submission acceptable for publication.

Experimental design

no comment

Validity of the findings

no comment

Additional comments

Recommendation: accept for publication.

Reviewer 3 ·

Basic reporting

The manuscript is clear indeed and the article's structure is proper. Moreover, this new version on the manuscript has been improved in terms of background and context, addressing many of the reviewers' previous suggestions and concerns.

Experimental design

No comment

Validity of the findings

The changes made to the manuscript in this new version contribute to the solidness of the conclusions. The article is now suitable for publication.